## RESEARCH ARTICLE

# Examining the availability/findability of stimuli employed in social media and body image research

David Smailes[1]*, Arnela Aleksandra[1], Megan Coakley[1], Susan Mair[2], Joe Ventress[1]

**1** Department of Psychology, Northumbria University, Newcastle upon Tyne, England, **2** Content Delivery and Copyright, Student, Library and Academic Services, Northumbria University, England

* david.smailes@northumbria.ac.uk

## Abstract

Concerns over the trustworthiness of the research findings generated in Psychology (as well as other disciplines) has led to calls for the adoption of practices that make research more open, transparent, and reproducible. One of these practices is the open sharing of research materials, such as task stimuli. There is some evidence that, generally, the uptake of this practice has been slow in Psychology. The aim of this study was to examine the availability/findability of the stimuli used in a sample of papers that investigated the effect of exposure to images from social media on participants' body image, as this may be a field where progress in the open sharing of task stimuli may be especially slow. We coded the method sections of 38 studies (published across 36 articles from 2012 to 2021) in terms of the availability/findability of the images they employed and found that in only two articles were we able to fully access task stimuli. We also found no evidence that the sharing of images used as task stimuli had increased over time. We discuss likely reasons for this reticence to share task stimuli in this field, the impact this has on reproducibility, replicability, and research waste, and ways in which this issue can be addressed. All study materials and data are available at doi.org/10.17605/osf.io/wpvst.

### Examining the availability/findability of stimuli employed in social media and body image research

Since 2012, and the start of the 'replication crisis' [1], Psychology researchers (as well as researchers in other disciplines, e.g., biology; see [2]) have increasingly engaged in a set of 'open research' practices. 'Open research' practices include the pre-registration of predictions and analysis plans (e.g., [3]), the use of open-source software (e.g., [4]), and the open sharing of data (e.g., [5]). By adopting these practices, individuals aim to increase the transparency and rigor of their research. As a result, the likelihood that they will generate robust, replicable findings should be increased.

**Data availability statement:** The data for this study are available at doi.org/10.17605/osf.io/wpvst.

**Funding:** The author(s) received no specific funding for this work.

**Competing interests:** All other authors have declared that no competing interests exist.

A further key 'open research' practice is the sharing of research materials, such as questionnaires and tasks. Reproducing others' work is a cornerstone of science [6] and researchers who openly share their research materials facilitate this core process [7]. However, it appears that the norm across many disciplines is to not openly share research materials. For example, in a recent study that sampled 250 articles published in Psychology journals between 2014 and 2017, Hardwicke et al. [8] reported that research materials were shared in only 14% of those articles. Similarly, in an analogous study where 250 social science articles (i.e., articles from Psychology, but also from Geography, Economics, and Political Science, for example) published between 2014 and 2017 were examined, 11% were rated as openly sharing their research materials [9].

One particular area within Psychology where the sharing of research materials could be considered to be especially important is in research that examines the impact of images from social media on body image. This evidence base typically examines how exposure to images from social media that, for example, promote the 'thin ideal' (e.g., [10]) or that serve as 'fitspiration' (e.g., [11]) affect participants' body image (or related constructs) versus exposure to control images (such as nature scenery). The images used in these studies are clearly very important to the design of the study, as their content should determine the magnitude of the intended manipulation, and so it could be considered especially important that researchers working in this area share the images they have used as task stimuli. However, the experience of two of the authors (AA and DS) in designing a study that examined the effect of exposure to images from social media that promote the 'thin ideal' on women aged 18–22 years versus women aged 31–45 years (see: doi.org/10.17605/osf.io/zfjqh), suggested that the sharing of stimuli was very uncommon in this area.

Given this experience, the present study aimed to investigate this issue systematically, by examining how available (or findable) images used as task stimuli in research on social media and body image were. In addition, given that there is some indication that open research practices are becoming more common over time (e.g., [12]), we tested the possibility that there would be an association between year-of-publication and open sharing of the images used as task stimuli.

## Method

### Sample of articles

Rather than performing a novel literature search, we used all 36 articles synthesized as part of de Valle et al.'s [13] meta-analysis, which examined the impact of exposure to images from social media on body image, as our sample of our to-be-coded articles. Our sample of papers is, therefore, made up of the papers listed in Table A1 of the Supplementary Materials to de Valle et al.'s study. The sample of papers is also available at https://osf.io/wfejz.

We took this approach (re-using the corpus of articles generated by de Valle et al. [13] rather than performing a novel literature search) in an effort to reduce research waste [14]. That is, our judgement was that if we performed a similar literature search to the one conducted by de Valle et al. [13], our search would generate only a few

more additional eligible studies (given that we began this project in January 2023, and de Valle et al. completed their searches in February 2021) and that the benefit of identifying these additional search results would not outweigh the cost (in terms of researcher time) of performing a novel literature search.

The 36 articles reported findings from 38 studies, and were published between 2012 and 2021. The sample sizes of the studies varied from 47 to 501, with the majority of studies involving exclusively participants who self-reported their gender as female. Each study employed an experimental method which involved exposing at least one group of participants to images that were taken from social media websites (such as Facebook or Instagram), or were made to look as if they were taken from a social media website (with the images purchased from companies such as ShutterStock, a stock photography provider, and then edited). In some studies (e.g., [15]), the effect of being exposed to images from social media versus, for example, being exposed to images of scenes of nature on participants' body image/satisfaction was compared. In other studies (e.g., [16]), the effect of being exposed to one type of images from social media (e.g., those that promoted the 'thin ideal') versus being exposed to other types of images from social media (e.g., those that parodied the 'thin ideal') on participants' body image/satisfaction was compared.

## Coding of studies

We drafted a coding system based on our past experiences of reading papers that investigated the effect of exposure to images from social media on participants' body image. One author (JV) piloted the coding system on all 36 studies. After this pilot, we refined the coding system (e.g., making some language more precise, revising some examples), and then all studies were reviewed and coded by two authors (DS and MC) using this revised coding scheme, to establish inter-rater reliability. Where there were disagreements in the coding of these two raters, agreement on the most appropriate code was achieved through discussion. Coding involved reading the full-text manuscripts of the to-be-coded articles, as well as accessing supplementary materials or online repositories whenever needed.

The coding scheme – which is available at https://osf.io/7nxja – involved reviewing studies in terms of four codes, with the highest code (the fourth code) reflecting the most 'open' practices in terms of making task stimuli available, findable, or reproducible. Up to the point of researchers sharing the stimuli they employed in a completely open or reproducible manner (e.g., sharing the images used, or providing URLs where the images could be found), we also tried to code for whether the information provided made the work more or less reproducible. We did this by considering how precisely the study described the possible set of images someone could use if they were trying to reproduce the task stimuli. So, for example, we considered whether a study gave only examples of the kinds of social media accounts they took images from, or whether they explained exactly which social media accounts they took images from. We considered the latter as being a more precise description of the images used and, therefore, as increasing the likelihood that a researcher would be able to reproduce the original task stimuli. Or, if a study reported using images that were associated with a specific hashtag, we considered how many image results that hashtag was associated with. A study that used a hashtag associated with a small number of images to source task stimuli was considered to have provided a more precise description of the images used than a study that used a hashtag associated with a large number of images to source task stimuli. Our coding system often referred to Instagram, to metrics relevant to Instagram (such as number of followers, or number of posts from an account), and to social media accounts of women, as the majority of the to-be-coded studies focused on the effects of images of women taken from Instagram on participants who self-reported their gender as female.

The first code assessed whether the authors provided a generic description of the images used. For example, did they report something like "images used were taken from the accounts of young female celebrities and depicted those celebrities in revealing clothing/as being extremely thin/being extremely muscular"?. This was scored 'yes' versus 'no'.

The second code assessed whether the authors provided example images, examples of accounts used, and/or a widely-used hashtag that was used to identify images. For example, did they provide a small set of example images (here we set a threshold of less than 50% of the images used), did they report that images were taken from the accounts of

celebrities in an imprecise way (e.g., stating that images were "taken from the accounts of celebrities such as Kendall Jenner and Ariana Grande"), and/or did they provide a hashtag, which when we searched for that hashtag resulted in more than 1,995 image results. We set this threshold of 1,995 images on the following basis. We accessed a list of the 50 most-followed accounts on Instagram (from https://en.wikipedia.org/wiki/List_of_most-followed_Instagram_accounts; accessed on 21 February 2023). From that list, we identified the accounts that belonged to women under the age of 50 years. There were 26 of these, and we calculated the median number of posts from those 26 accounts. The median was 1,994.5, which we rounded up to 1,995. Our logic here, then, was that providing a hashtag that resulted in more than 1,995 images was less precise than reporting that images were taken from 'an average' highly-followed, young, female celebrity account on Instagram. Again, this was scored 'yes' versus 'no'.

The third code assessed whether the authors provided the majority of the images used, exhaustive information about accounts used, and/or a not-very-widely-used hashtag that was used to identify images. For example, did they provide a large set of example images (50% or more of the images used), did they report that images used were taken from the accounts of Kendall Jenner, Ariana Grande, Harry Styles, and Chris Hemsworth (and these accounts alone), and/or did they report that images used were sourced using a specific hashtag, which when we searched for that hashtag resulted in no more than 1,995 image results? Our logic for using this threshold of 1,995 images was the same as above. Here, where researchers reported using a hashtag that resulted in no more than 1,995 images, this was as precise, or more precise, as reporting that images were taken from 'an average' highly-followed, young, female celebrity account on Instagram. Again, this was scored 'yes' versus 'no'.

The fourth code assessed whether the authors provided URLs for the images used and/or provided all of the images used. These images/URLs could be provided in the manuscript, in supplementary materials, or at some other online repository (provided we were able to access the repository). Again, this was scored 'yes' versus 'no'.

When coding articles, we coded multiple studies within the same article separately, and so these appear as separate rows in the spreadsheet at https://osf.io/vzhj9. Some studies involved multiple sets of images from social media, which were described separately in the article (e.g., images from the social media accounts of men versus images from the social media accounts of women; images from social media that promoted the 'thin ideal' versus images from social media that parodied the 'thin ideal'). Again, these appear as separate rows in the spreadsheet at https://osf.io/vzhj9. Where an article reported that stimuli was available upon request, we did not take this into account, as we assumed that – as in the case for data-sharing where data is reported to be available on request (e.g., [4]) – the materials would not be easily/meaningfully available through this route.

To examine inter-rater reliability, we took the 'highest' code that a rater had scored 'yes' and tested agreement for this. For 88% of scores given (36 out of 41), there was agreement between the two raters. We used Cohen's kappa as our measure of agreement and found 'substantial' levels of agreement (kappa = 0.80). In addition, we examined level of agreement code-by-code (i.e., Did DS and MC both score 'yes' or both score 'no' for Code 1? Did DS and MC both score 'yes' or both score 'no' for Code 2? And so on). For Code 1, there was perfect agreement (both raters scored 'yes' for all studies). For Code 2, we found 'substantial' levels of agreement (same scores given for 39 of 41 studies; kappa = 0.88). For Code 3 (same scores given for 38 of 41 studies; kappa = 0.78) and Code 4 (same scores given for 40 of 41 studies; kappa = 0.79), we found 'moderate' levels of agreement.

## Results

### Availability of task stimuli

We rated two studies (reported across two publications; [17,18]) as sharing images/providing links to images in a way that made their task stimuli fully reproducible, as they provided all of the images they had used as stimuli. In one of these articles, images used as task stimuli were shared in an appendix to the full-text. In the other article, images were shared at flickr.com. We rated four studies (reported across four publications) as providing the majority of the images used,

exhaustive information about accounts used, and/or a not-very-widely-used hashtag that was used to identify images. We rated 23 studies (reported across 22 publications) as providing a small sample of example images, examples of accounts used, and/or a widely-used hashtag that was used to identify images. Finally, we rated 12 studies (reported across 10 publications) as describing the images used in a way that was least helpful in reproducing their task stimuli (i.e., they gave only generic descriptions of the kinds of images employed as stimuli. For example, "Eighteen images from high-popularity female influencers (defined for this purpose as users with>100k followers; 101k-1.2 M), were chosen on the criteria of high-quality (clear and professional appearing) imagery and relative obscurity (i.e., accounts from European and US influencers who are not typically internationally renowned/recognisable, based on pilot data from three Australian female participants). The final profiles depicted thin, attractive, white female influencers (aged ~20−30) with a combination of close up and full-body photos featuring the influencer from a variety of angles (i.e., both facing and looking away from the camera). The images depicted the influencers in artistic, lifestyle, and travel selfies. Influencers were fully clothed in summer attire (except one swimsuit photograph depicting the subject's back) and most photos appeared to be staged/posed rather than spontaneous)"). Study-by-study ratings are available at https://osf.io/vzhj9.

## Association between Availability of Stimuli and Year-of-Publication

To test the possibility that the authors of studies published more recently were more likely to have engaged in more open research practices, we ran an exploratory analysis where we correlated year-of-publication with the 'highest' code for which a 'yes' score was given. Analysis was performed in Jamovi (version 2.3.21; [19,20]). We found that there was a non-significant, positive correlation between year-of-publication and highest code scored yes (rho = .12, p = .60).

## Discussion

The aim of this study was to examine the availability/findability of the images used as task stimuli in research that tested the effect that exposure to images from social media have on participants' body image (and related constructs). We found that very few of the articles we coded made the images employed fully available. Many more articles gave only broad, imprecise descriptions of the images they used. Thus, it seems that in experimental research on the impact of social media on body image, there has been little adoption of open sharing of study materials.

These findings are consistent with the results of past research that has examined the availability of tasks/task stimuli in Psychology research. For example, Hardwicke et al. [8] reported that research materials were shared in only 14% of a random sample of Psychology articles published between 2014 and 2017. Similarly, across a random sample of 250 articles from the social sciences published between 2014 and 2017, only 11% openly shared their research materials [9]. Thus, the pattern we have reported here, of the norm in social media and body image research being a failure to share research materials, is also the norm in other areas of Psychology and in other social science disciplines.

The lack of open sharing of the images used as task stimuli in social media and body image research likely has several implications or consequences. First, this makes reproducing others' work much more difficult, as without certainty around what stimuli were used in a study researchers cannot be confident that they are creating a task that is equivalent to the one used in the original study. Given that reproducing others' research and attempting to replicate their findings is a cornerstone of the scientific method [6], it is clearly a substantial problem if researchers are not able to do so.

Related to this is a second consequence: the ability of researchers to 'successfully' replicate others' work (i.e., to find the same effects that have been reported in previous studies) is probably reduced by the failure to openly share task stimuli. This is because where a researcher cannot precisely re-create the task used in an original study, and so must source a different set of stimuli when developing their own version of the task for a subsequent study, the magnitude of the effect elicited by the task in the subsequent study will almost certainly differ from the magnitude of the effect elicited by the task in the original study. And so, across studies that use different sets of stimuli, we should expect effect sizes to vary beyond sampling error, possibly to the extent that some studies will find significant effects on body image and other studies will

not. Thus, what appears to be inconsistency within an evidence base, may simply be a result of the use of variation in stimuli employed. To some extent, variation in the stimuli used across studies can strengthen an evidence base as it shows whether an effect is generalizable. And, as de Valle et al. [13] noted this was the case in their meta-analysis, where effects were similar across studies that employed different types of images (e.g., images that promote a traditional 'thin ideal' image versus images that promote specific forms of 'fitspiration'). However, if materials were shared openly, it would be easier for researchers to examine the effects of the use of different images in a systematic manner (as Yarkoni [21] recommended).

The third consequence of a lack of sharing of images used as task stimuli in social media and body image research is one of research waste [14]. Without straightforward access to the task stimuli that others have used previously, researchers will unnecessarily spend a (presumably) substantial amount of time sourcing their own stimuli, building a task using this stimuli, and (ideally) piloting this task to ensure it works as intended. At present, there are no good estimates of how much time researchers spend unnecessarily developing tasks that could have been shared openly. It would be useful for future research to provide these kind of estimates, as has been done for estimating the cost of conducting a meta-analysis (around $141,000; [22]), and the number of 'redundant' meta-analyses that are published (possibly more than 50% of all published meta-analyses; [23]).

It is likely that there are a number of systemic factors that have contributed to the lack of sharing of images used in social media and body image research. For older studies in our sample, it is likely that sharing task stimuli was both not something that researchers were actively encouraged to do and was something that was relatively complex to achieve. For example, the Open Science Framework, which many researchers now use to share data and research materials [24], was only established in 2013. Prior to its existence, researchers may not have had a good understanding of which online repositories would be effective places to share study materials from. More recently, it is likely that a simple lack of resources/time-pressure discourage researchers from sharing task stimuli.

However, another issue may be that researchers are hesitant to share tasks that feature stimuli taken from social media because of concerns about privacy or copyright. That is, they may worry that if they develop a task that contains images from a celebrity's social media account and then share that task freely online, they may be violating copyright law. Such concerns are warranted in some contexts (see [25]), but will not be relevant for many researchers, as we explain below.

Concerns about privacy are very much warranted where researchers have used images (or other types of content, such as text) posted to 'private' accounts, or content that the original poster intended to be private to some extent. See, for example, Morant et al. [26], where content from Twitter was used by researchers in a way that was upsetting for the original Twitter users, who felt they had engaged in private conversations, which should not have been used as data by the researchers without consent being obtained. In these contexts, researchers should request permission from the account holder to use the content they have posted for research purposes. If this is provided, then researchers should discuss with the account holder whether they also consent to that content being shared openly with other researchers (e.g., via the Open Science Framework, or in supplementary materials to an article), as it is possible that an account holder may consent to content from their posts being used in a single research project, but may not consent to that content being widely shared.

In contrast, concerns about copyright law are less warranted where researchers have used images posted to 'public' social media accounts, which are easily discoverable by anyone accessing a social media platform such as Instagram, Facebook, Twitter, or TikTok. Typically these images can be used under the doctrine of 'fair use' (see help.instagram.com/116455299019699?helpref=faq_content and https://copyright.gov/fair-use/), as long as researchers include a citation to accompany the content they have used/shared. A good example of this is Lalancette and Raynauld's [27] article, which analysed the use of online images in 'celebrity politics' in Canada, and re-produced 11 images from Canadian Prime Minister Justin Trudeau's Instagram account, with each image accompanied by an appropriate citation. Similarly, as part of a research project conducted by AA and DS, we posted PowerPoint

files containing the images we employed as stimuli on the Open Science Framework (see https://osf.io/egamu and https://osf.io/yce9z) in a way that does not violate copyright law.

Researchers can raise concerns they have around privacy and copyright with copyright librarians at their institutions, who should be able to provide clear guidance about what best practice is in terms of re-using and sharing content taken from social media accounts. If this advice is not available to a researcher, and they wish to err on the side of caution, then posting the URLs where the images used can be found, as well as a detailed description of the images, is another way in which researchers can make their stimuli available to others (see https://osf.io/y796h and https://osf.io/g7xem for examples of this). However, it should be noted that there are limitations to this approach of sharing URLs, as they may be affected by 'link rot' [28], where once-usable links become unavailable over time. This is especially likely to be a problem when trying to provide links to social media accounts, as it is possible to automatically delete posts from social media, once a post is a certain 'age' or so that an account does not contain more than a specific number of posts (see tweetdeleter.com/features/auto-delete-tweets and aischedul.com/delete-instagram-posts-automatically-after-publishing/). Thus, best practice likely involves posting both the URLs where the images used as task stimuli can be found, as well as re-productions of the images used, accompanied by appropriate citations.

It is, therefore, possible for researchers – at least in most cases – to share the images from social media that they have used as task stimuli, and there are good reasons (related to increasing reproducibility, increasing replicability, and reducing research waste) to do so. However, here we found no evidence that open sharing of stimuli had increased over time, which suggests that adoption of this practice will not occur without specific interventions. Interventions that may increase the open sharing of stimuli by researchers working on social media and body image include journals implementing policies that mandate the sharing of research materials and/or journals awarding 'badges' that acknowledge articles where the authors have shared research materials. Journal policies mandating data-sharing (or the inclusion of a statement explaining why data cannot be shared) have been shown to be associated with a substantial increase in that practice, although it should be noted that data was often shared in sub-optimal ways [29]. The awarding of badges by journals for articles that have adopted 'open research' practices, such as the pre-registration of analysis plans and the sharing of analysis code, has also been associated with an increase in the adoption of these practices [30]. The *International Journal of Eating Disorders* has recently introduced a 'badges' system and new editorial policies around open science [7], and research that examines the impact these interventions have on the open sharing of research materials on social media and body image research published in that journal versus social media and body image research published elsewhere will be of interest.

This study suffered from several limitations. First, the sample of articles we coded was relatively small in comparison to other studies that have examined 'open research' practices (e.g., samples of 250 papers were used in [8] and [9]). That being said, this study focused on a narrower subset of articles, and so this smaller sample of papers is almost inevitable. In addition, others (e.g., [31]) have used similar (even smaller) samples of articles to investigate the kinds of questions we have examined here. Second, because we employed a sample of papers generated by researchers who conducted a meta-analysis in 2021, the articles we reviewed do not reflect practices researchers have engaged in 2022–2025. Given that we did not find an association between year-of-publication and increased sharing of images used as task stimuli, we would be surprised if there had been a substantial change in this practice in 2022 and 2023. Although, as we noted above, the *International Journal of Eating Disorders* has recently implemented new policies and incentives around engaging in 'open science' practices, and future research should examine if sharing of images used as task stimuli becomes more common in social media and body image research through the mid-2020s. Third, we developed our coding system based on our past experience of reading publications in this field, rather than using coding systems developed by other researchers. This may have introduced some bias in, for example, the codes generated. Fourth, the concerns we have raised about open sharing of materials would be less of a problem if, when contacted, the authors of a publication were able to straightforwardly share (e.g., via private email correspondence) the materials they had used. We took the approach of assuming that contacting authors would have been unproductive, and we think that this is a fair assumption based on,

for example, data that around 85% of authors do not respond to requests to share data [32]. However, it is possible that authors would be more willing to discuss sharing of materials than data. Future research that examines how often authors of social media and body image research are willing/able to share testing materials when requested would be valuable.

Finally, another limitation caused by our use of a sample of papers generated by other researchers when they were conducting a meta-analysis is that the value of our study depends, at least to some extent, on the comprehensiveness of their search strategy and the rigor with which it was carried out. Our judgment was that the literature search conducted by de Valle et al. [13] was an effective one, in that the terms employed were broad enough so that relevant studies were discovered and in that several of the criteria by which systematic reviews are evaluated (using more than one database to search for publications, using more than one researcher to make decisions about including/excluding publications, using more than one researcher to extract data from publications; [33]) were met. However, it is true that the limitations that applied to de Valle et al.'s search strategy (e.g., excluding studies that were not written in English) also apply to the current study.

In summary, this study aimed to examine the availability/findability of images used as task stimuli in social media and body image research. In line with research that has examined similar questions in other disciplines, we found that open sharing of task stimuli was rare. Future research that examines whether this changes throughout the 2020s, as open research practices become incentivized by a growing number of journals, will be valuable.

## Author contributions

**Conceptualization:** David Smailes, Arnela Aleksandra.

**Formal analysis:** David Smailes.

**Investigation:** David Smailes, Arnela Aleksandra, Megan Coakley, Joe Ventress.

**Methodology:** David Smailes.

**Writing – original draft:** David Smailes.

**Writing – review & editing:** Arnela Aleksandra, Megan Coakley, Susan Mair, Joe Ventress.

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
