## [Decision Letter · Decision Letter 0]

17 Feb 2025

PONE-D-24-48265Examining the availability/findability of stimuli employed in social media and body image researchPLOS ONE

Dear Dr. Smailes,

Thank you for submitting your manuscript to PLOS ONE. After careful consideration, we feel that it has merit but does not fully meet PLOS ONE’s publication criteria as it currently stands. Therefore, we invite you to submit a revised version of the manuscript that addresses the points raised during the review process.

We look forward to receiving your revised manuscript.

Kind regards,

Tyler Horan

Academic Editor

PLOS ONE

1. Please ensure that your manuscript meets PLOS ONE's style requirements, including those for file naming. The PLOS ONE style templates can be found at https://journals.plos.org/plosone/s/file?id=wjVg/PLOSOne_formatting_sample_main_body.pdf and https://journals.plos.org/plosone/s/file?id=ba62/PLOSOne_formatting_sample_title_authors_affiliations.pdf.

2. Please match your authorship list in your manuscript file and in the system.

3. Your abstract cannot contain citations. Please only include citations in the body text of the manuscript, and ensure that they remain in ascending numerical order on first mention.

[None]. 

Additional Editor Comments:

Based on the reviews submitted, I recommend that the author revise the manuscript to take into account the recommendations provided by both reviewers. I agree with their assessment and welcome a revised version that incorporates the feedback.

Reviewers' comments:

Reviewer's Responses to Questions

**Comments to the Author**

1. Is the manuscript technically sound, and do the data support the conclusions?

Reviewer #1: Yes

Reviewer #2: Yes

2. Has the statistical analysis been performed appropriately and rigorously? 

Reviewer #1: Yes

Reviewer #2: Yes

3. Have the authors made all data underlying the findings in their manuscript fully available?

Reviewer #1: Yes

Reviewer #2: Yes

4. Is the manuscript presented in an intelligible fashion and written in standard English?

Reviewer #1: Yes

Reviewer #2: Yes

5. Review Comments to the Author

Reviewer #1: I appreciated this study which was pretty direct an interesting. I have just a few thoughts.

I don’t you need to refer to a “so called” replication crisis in the first line. I think that can communicate skepticism, which I don’t think was your intent. There’s not too much doubt remaining at this point that social science experienced a replication crisis.

I appreciated the discussion of the issue on copyright. I’m not a lawyer but I’m not sure that posting something publicly invalidates copyright. For instance a song or film publicly released doesn’t lose copyright protection. I do like the suggestion about providing links to material (with the caveats the authors mention about the decay of those links being well-taken).

I’d be curious regarding a further point of data…if the authors tried reaching out to article authors for their full material…what would be their reaching out success? That might be one little data point that would be interesting to add. Open materials may be a bigger/lesser issue depending on how good researchers are about responding to requests for materials.

Reviewer #2: This study examined the public availability of study stimuli used in social media and body image research from the de Valle et al. (2021) review. Results showed that few studies provided open access to the stimuli used. This is an important issue to raise. However, it would be more helpful if the paper sought to understand why these practices have not been adopted. For example, researchers could be surveyed to better understand the barriers to open sharing of their stimuli. The paper would be more influential if it helped overcome barriers to allow for more open science practices. Below, I suggest ways to improve the manuscript.

1) The authors imply that the lack of stimuli sharing is due to poor open science practices. My studies were included in the review, along with that of my colleagues. I have not shared the stimuli because my ethics committee have not allowed me to do so, despite my highlighting the public nature of the posts. They argue that I do not own or have permission to reproduce the images. This is a common occurrence across many institutions. The authors recommend contacting copyright librarians. However, to reduce research waste, it would be helpful for the authors to mention specific resources or policies that researchers could provide to their ethics committees to argue against privacy concerns and to advocate for the open sharing of public social media posts. This would help reduce barriers to open sharing of stimuli within the field.

2) I think it should also be acknowledged that similar effects are often found within the literature despite using different stimuli, as highlighted in the de Valle et al. (2021) review. Given the research aims to determine the effect of social media images on body image, I think it is important to ensure that the effects generalise to other images. I do not see this as research waste but rather as an important step to ensure the findings are not specific to one set of images.

3) The field moves quickly, and many studies have been published on this topic since 2021. I was surprised to see researchers’ time be mentioned as a reason for not doing a search for more recent literature. This does not seem to be a strong justification given that time since publication is examined as a moderator and it has been four years since the data were searched.

4) On Page 8, it states that studies in which data were available on request were not taken into account because it was assumed that those data would not be easily available. This assumption does not seem fair. If stimuli can be accessed by emailing the researchers, that would overcome issues related to reproducibility, replicability, and research waste.

6. PLOS authors have the option to publish the peer review history of their article (what does this mean? ). If published, this will include your full peer review and any attached files.

**Do you want your identity to be public for this peer review?** For information about this choice, including consent withdrawal, please see our Privacy Policy .

Reviewer #1: **Yes: ** Christopher Ferguson

Reviewer #2: No

---

## [Author Response · Author response to Decision Letter 1]

21 Mar 2025

Thank you for the helpful comments. Please find our response to the reviewers’ comments below.

Reviewer #1: Peer Review

Comment 01 = I don’t you need to refer to a “so called” replication crisis in the first line. I think that can communicate skepticism, which I don’t think was your intent. There’s not too much doubt remaining at this point that social science experienced a replication crisis.

Response 01 = Thanks very much for this comment. We have deleted “so-called” so that the sentence (Line 41) now reads “Since 2012, and the start of the ‘replication crisis’ (Lillenfield & Strother, 2020)…”.

Comment 02 = I appreciated the discussion of the issue on copyright. I’m not a lawyer but I’m not sure that posting something publicly invalidates copyright. For instance a song or film publicly released doesn’t lose copyright protection. I do like the suggestion about providing links to material (with the caveats the authors mention about the decay of those links being well-taken).

Response 02 = Apologies that we weren’t clearer in the manuscript. Copyright isn’t invalidated when images are publicly posted on platforms such as Instagram. But, under a ‘fair use’ doctrine, images from social media can be re-used by others in ways that don’t infringe the author’s copyright (and the ways we describe re-using images from social media would [we are confident] be classed as ‘fair use’). We have included specific reference to the ‘fair use’ doctrine in the revised manuscript (Lines 287-289), and have included links to two websites - help.instagram.com/116455299019699?helpref=faq_content and https://copyright.gov/fair-use/ - that provide more detailed information. We hope that these additions provide extra context/information that is helpful.

Comment 03 = I’d be curious regarding a further point of data…if the authors tried reaching out to article authors for their full material…what would be their reaching out success? That might be one little data point that would be interesting to add. Open materials may be a bigger/lesser issue depending on how good researchers are about responding to requests for materials.

Response 03 = We also think that this would be a very interesting research question but we think we should raise two points. First, sharing materials via email correspondence, versus ‘open sharing’ via a public repository, is less effective (even with the best of intentions) as authors’ email addresses can change over time, and when this happens, tracking down an author can either be time-consuming, difficult, or impossible. For example, Gabelica et al. (2022) reported that around 5% of the email addresses they contacted no longer worked, when they requested data from recently published papers (articles were all published in January 2019; Gabelica et al. was accepted for publication in May 2022). Presumably this problem will become worse over time. Second, we feel that this question is a little bit beyond the scope of this specific manuscript. As we note again below, we think that a larger-scale, follow-up study that examined the availability/findability of stimuli employed in social media and body image research (a) from 2021 to end-of-2025, (b) by using the method employed here, and (c) by also testing how effective requests for copies of materials were (as suggested by Reviewer #1 here, and by Reviewer #2 below) would be extremely useful.

Reviewer #2: Peer Review

Comment 04 = However, it would be more helpful if the paper sought to understand why these practices have not been adopted. For example, researchers could be surveyed to better understand the barriers to open sharing of their stimuli. The paper would be more influential if it helped overcome barriers to allow for more open science practices. Below, I suggest ways to improve the manuscript.

Response 04a = We agree that examining why materials are rarely shared would be both interesting and useful. However, we think that a project formally examining this (e.g., via a survey, as suggested) is beyond the scope of this manuscript (ideally this is something we would be keen to examine in a larger-scale, follow-up study, which we note elsewhere).

Response 04b = We also agree that manuscripts which identify ways to address problems are more valuable than manuscripts that solely identify the existence of a problem. We intended that our manuscript would offer some solutions to what we assumed was a key barrier – concerns about copyright infringement – by providing some advice that might reduce researchers’ concerns about copyright infringement. However, we accept that the previous version of the manuscript didn’t provide sufficient information/context about this issue, and so we have added links (Lines 287-289) to two websites - help.instagram.com/116455299019699?helpref=faq_content and https://copyright.gov/fair-use/ - that provide more detailed information. We hope that these additions provide extra context/information that is helpful.

Comment 05 = The authors imply that the lack of stimuli sharing is due to poor open science practices. My studies were included in the review, along with that of my colleagues. I have not shared the stimuli because my ethics committee have not allowed me to do so, despite my highlighting the public nature of the posts. They argue that I do not own or have permission to reproduce the images. This is a common occurrence across many institutions. The authors recommend contacting copyright librarians. However, to reduce research waste, it would be helpful for the authors to mention specific resources or policies that researchers could provide to their ethics committees to argue against privacy concerns and to advocate for the open sharing of public social media posts. This would help reduce barriers to open sharing of stimuli within the field.

Response 05a = Apologies, we did not intend to imply that failing to share materials openly was poor practice, or the fault of specific researchers (we have added the word ‘systemic’ in Line 261 to emphasize that we think this is a ‘system-problem’ rather than the fault of individual researchers). As we noted in the original manuscript, we expect that this happens for a variety of understandable reasons such as a lack of time/resource, the absence of appropriate platforms through which stimuli can be shared, and concerns about copyright/privacy.

Response 5b = We agree that the manuscript would have been stronger if we had provided links to resources that offer clear advice about the re-use of images from social media. Thus, in the revised manuscript, we have included specific reference to the ‘fair use’ doctrine (Lines 287-289), and have included links to two websites - help.instagram.com/116455299019699?helpref=faq_content and https://copyright.gov/fair-use/ - that provide more detailed information about this. We hope that these additions provide extra context/information that is helpful in, for example, discussions with ethics committees.

Comment 06 = I think it should also be acknowledged that similar effects are often found within the literature despite using different stimuli, as highlighted in the de Valle et al. (2021) review. Given the research aims to determine the effect of social media images on body image, I think it is important to ensure that the effects generalise to other images. I do not see this as research waste but rather as an important step to ensure the findings are not specific to one set of images.

Response 06 = This is a very valid point. Thank you for raising it. We have noted in the revised manuscript that this (consistent effects across different stimuli) was found in de Valle et al.’s meta-analysis. However, we note that open sharing of materials would make this kind of analysis more straightforward for researchers to conduct (Line 246-250).

Comment 07 = The field moves quickly, and many studies have been published on this topic since 2021. I was surprised to see researchers’ time be mentioned as a reason for not doing a search for more recent literature. This does not seem to be a strong justification given that time since publication is examined as a moderator and it has been four years since the data were searched.

Response 07 = We also think that examining whether open sharing of materials has become more common post-2021 would be a very interesting research question. Apologies for not being clearer in the manuscript, but it is not that we think assessing practices post-2021 is not worth our time. Our argument is that the corpus of papers identified by de Valle et al. (2021) enabled us to run the present project with little resource/at low cost and that this was very valuable. In part this is because the results we reported should allow us to obtain more resource to run the kind of project Reviewer #2 suggests in their comments. That is, as we note again below, we think that a larger-scale, follow-up study that examined the availability/findability of stimuli employed in social media and body image research (a) from 2021 to end-of-2025, (b) by using the method employed here, and (c) by also testing how effective requests for copies of materials were would be extremely useful.

Comment 08 = On Page 8, it states that studies in which data were available on request were not taken into account because it was assumed that those data would not be easily available. This assumption does not seem fair. If stimuli can be accessed by emailing the researchers, that would overcome issues related to reproducibility, replicability, and research waste.

Response 08 = We also think that this would be a very interesting research question and the reviewer is clearly correct that this would, to some extent, reduce the impact of the problem we identified in the manuscript. However, we feel that this is beyond the scope of this specific manuscript. As we noted above, we think that a larger-scale, follow-up study that examined the availability/findability of stimuli employed in social media and body image research (a) from 2021 to end-of-2025, (b) by using the method employed here, and (c) by also testing how effective requests for copies of materials were (as suggested by Reviewer #2 here, and by Reviewer #1 above) would be extremely useful. We’d very happily discuss the possibility of developing that sort of study with Reviewer #2 at some point in the future, if they were keen to collaborate on that kind of project.

We have noted in the Discussion section (Line 343-351) that this is a limitation of the present study, we note that we think our assumption that contacting authors would likely have been unproductive (given that, for example, Gabelica et al. [2022] reported that 85.83% of authors contacted re: sharing their data did not respond), and that future research should examine this issue re: social media and body image research.

Novel References

Gabelica, M., Bojčić, R., & Puljak, L. (2022). Many researchers were not compliant with their published data sharing statement: a mixed-methods study. Journal of Clinical Epidemiology, 150, 33-41.

---

## [Decision Letter · Decision Letter 1]

28 Apr 2025

Examining the availability/findability of stimuli employed in social media and body image research

PONE-D-24-48265R1

Dear Dr. Smailes,

We’re pleased to inform you that your manuscript has been judged scientifically suitable for publication and will be formally accepted for publication once it meets all outstanding technical requirements.

Kind regards,

Tyler Horan

Academic Editor

PLOS ONE

Additional Editor Comments (optional):

Reviewers' comments:

Reviewer's Responses to Questions

**Comments to the Author**

1. If the authors have adequately addressed your comments raised in a previous round of review and you feel that this manuscript is now acceptable for publication, you may indicate that here to bypass the “Comments to the Author” section, enter your conflict of interest statement in the “Confidential to Editor” section, and submit your "Accept" recommendation.

Reviewer #1: All comments have been addressed

Reviewer #2: (No Response)

2. Is the manuscript technically sound, and do the data support the conclusions?

Reviewer #1: Yes

Reviewer #2: Yes

3. Has the statistical analysis been performed appropriately and rigorously? 

Reviewer #1: Yes

Reviewer #2: Yes

4. Have the authors made all data underlying the findings in their manuscript fully available?

Reviewer #1: Yes

Reviewer #2: Yes

5. Is the manuscript presented in an intelligible fashion and written in standard English?

Reviewer #1: Yes

Reviewer #2: Yes

6. Review Comments to the Author

Reviewer #1: (No Response)

Reviewer #2: I appreciate the resources provided by the authors regarding copyright laws. These will help researchers argue for the sharing of public posts in publications. The rest of my suggestions were deemed outside of the scope of the paper. I believe the study proposed as a next step would be a more helpful addition to the literature than the current paper. I have no further suggestions.

7. PLOS authors have the option to publish the peer review history of their article (what does this mean? ). If published, this will include your full peer review and any attached files.

**Do you want your identity to be public for this peer review?** For information about this choice, including consent withdrawal, please see our Privacy Policy .

Reviewer #1: **Yes: ** Christopher Ferguson

Reviewer #2: No

---

## [Editor Report · Acceptance letter]

PONE-D-24-48265R1

PLOS ONE

Dear Dr. Smailes,

I'm pleased to inform you that your manuscript has been deemed suitable for publication in PLOS ONE. Congratulations! Your manuscript is now being handed over to our production team.

Kind regards,

on behalf of

Dr. Tyler Horan

Academic Editor

PLOS ONE